# On the Almost Sure Convergence of the Stochastic Three Points Algorithm

**Taha EL BAKKALI EL KADI, Omar SAADI**
UM6P College of Computing
Benguerir, Morocco
{taha.elbakkali,omar.saadi}@um6p.ma

## Abstract

The stochastic three points (STP) algorithm is a derivative-free optimization technique designed for unconstrained optimization problems in $\mathbb{R}^d$. In this paper, we analyze this algorithm for three classes of functions: smooth functions that may lack convexity, smooth convex functions, and smooth functions that are strongly convex. Our work provides the first almost sure convergence results of the STP algorithm, alongside some convergence results in expectation. For the class of smooth functions, we establish that the best gradient iterate of the STP algorithm converges almost surely to zero at a rate of $o(1/T^{\frac{1}{2}-\epsilon})$ for any $\epsilon \in (0, \frac{1}{2})$, where $T$ is the number of iterations. Furthermore, within the same class of functions, we establish both almost sure convergence and convergence in expectation of the final gradient iterate towards zero. For the class of smooth convex functions, we establish that $f(\theta^T)$ converges to $\inf_{\theta \in \mathbb{R}^d} f(\theta)$ almost surely at a rate of $o(1/T^{1-\epsilon})$ for any $\epsilon \in (0, 1)$, and in expectation at a rate of $O(\frac{d}{T})$ where $d$ is the dimension of the space. Finally, for the class of smooth functions that are strongly convex, we establish that when step sizes are obtained by approximating the directional derivatives of the function, $f(\theta^T)$ converges to $\inf_{\theta \in \mathbb{R}^d} f(\theta)$ in expectation at a rate of $O((1 - \frac{\mu}{2\pi dL})^T)$, and almost surely at a rate of $o((1 - s\frac{\mu}{2\pi dL})^T)$ for any $s \in (0, 1)$, where $\mu$ and $L$ are the strong convexity and smoothness parameters of the function.

## 1 Introduction

We are interested in the minimization of a smooth function $f : \mathbb{R}^d \mapsto \mathbb{R}$:

$$\min_{\theta \in \mathbb{R}^d} f(\theta),$$

where we work within the constraint of not having access to the derivatives of $f$, relying exclusively on a function evaluation oracle. The methods used in this framework are called derivative-free methods or zeroth-order methods (Conn et al., 2009; Ghadimi & Lan, 2013; Nesterov & Spokoiny, 2017; Larson et al., 2019; Golovin et al., 2020; Bergou et al., 2020). They are increasingly embraced for solving many machine learning problems where obtaining gradient information is either impractical or computationally expensive, remaining crucial in applications such as generating adversarial attacks on deep neural network classifiers (Chen et al., 2017; Tu et al., 2019), reinforcement learning (Malik et al., 2019; Salimans et al., 2017), and hyperparameter tuning of ML models (Snoek et al., 2012; Turner et al., 2021). Therefore, exploring the theoretical properties of derivative-free methods is not only of theoretical interest but also crucial for practical applications.

Zeroth-order optimization methods can be divided into two main categories: direct search methods and gradient estimation methods. In direct search methods, the objective function is evaluated along a set of directions to guarantee descent by taking appropriate small step sizes. These directions can be either deterministic (Vicente, 2013) or stochastic (Golovin et al., 2020; Bergou et al., 2020). In contrast, gradient estimation methods approximate the gradient of the objective function using zeroth-order information to design approximate gradient methods (Nesterov & Spokoiny, 2017; Shamir, 2017).

A recent and noteworthy zeroth-order method is the Stochastic Three Points (STP) algorithm (see Algorithm 1), a directed search method with stochastic search directions, introduced by Bergou et al. (2020). The STP algorithm stands out among zeroth-order methods for its balance of simplicity and strong theoretical guarantees.

---

**Algorithm 1** Stochastic Three Points (STP)

---

1: **Input:** $\theta^1 \in \mathbb{R}^d$, step size sequence $\{\alpha_t\}_{t\geq 1} \in (0,\infty)^{\mathbb{N}^*}$, probability distribution $\mathcal{D}$ on $\mathbb{R}^d$.
2: **for** $t = 1, 2, \ldots$ **do**
3:      Generate a random vector $s_t \sim \mathcal{D}$,
4:      $\theta^{t+1} = \arg\min_{\theta \in \{\theta^t, \theta^t + \alpha_t s_t, \theta^t - \alpha_t s_t\}} f(\theta)$.
5: **end for**

---

Compared to deterministic directed search (DDS) methods, the worst-case complexity bounds for STP are similar; however, they differ in their dependence on the problem's dimensionality. For STP, the bounds increase linearly with the dimension (Bergou et al., 2020), whereas for DDS, they increase quadratically (Konečný & Richtárik, 2014; Vicente, 2013). Specifically, when the objective function is smooth, STP requires $O(d\epsilon^{-2})$ function evaluations to get a gradient with norm smaller than $\epsilon$, in expectation. For smooth, convex functions with a minimum and a bounded sublevel set, the complexity is $O(d\epsilon^{-1})$ to find an $\epsilon$-optimal solution. In the strongly convex case, this complexity reduces further to $O(d \log \epsilon^{-1})$. In all these cases, DDS methods exhibit analogous complexity bounds but with a quadratic dependence on $d$, i.e., $d^2$ instead of $d$. In comparison to directed search with stochastic directions, STP also matches the complexity bound derived by Gratton et al. (Gratton et al., 2015) for the smooth case, which is the only case they address in their work. In their approach, a decrease condition is imposed to determine whether to accept or reject a step based on a set of random directions. The Gradientless Descent (GLD) algorithm (Golovin et al., 2020) is another direct search method with stochastic directions. Golovin et al. show that an $\epsilon$-optimal solution can be found in $O(kQ \log(d) \log (R\epsilon^{-1}))$ for any monotone transform of a smooth and strongly convex function with latent dimension $k < d$, where the input dimension is $d$, $R$ is the diameter of the input space, and $Q$ is the condition number. When the monotone transformation is the identity and $k = d$, this complexity is higher than the one obtained for the STP algorithm by a factor of $\log(d)$. However, it is important to note that monotone transforms of smooth and strongly convex functions are not necessarily strongly convex.

Compared to approximate gradient methods, STP matches the complexity bounds of the random gradient-free (RGF) algorithm (Nesterov & Spokoiny, 2017) (see section 6) across the three cases: smooth non-convex, smooth convex, and smooth strongly convex. This matching in complexities is in terms of the accuracy $\epsilon$ and the dimensionality $d$.

In practical terms, for classical applications of zeroth-order methods, STP variants demonstrate strong performance when compared to state-of-the-art methods. For instance, in reinforcement learning and continuous control, specifically in the MuJoCo simulation suite (Todorov et al., 2012), STP with momentum (which, in expectation, achieves the same complexity bounds as standard STP, see Gorbunov et al. (2020)) outperforms methods like Augmented Random Search (ARS), Trust Region Policy Optimization (TRPO), and Natural Policy Gradient (NG) across environments such as Swimmer-v1, Hopper-v1, HalfCheetah-v1, and Ant-v1. Even in the more challenging Humanoid-v1 environment, STP with momentum achieves competitive results (Gorbunov et al., 2020). Additionally, in the context of generating adversarial attacks on deep neural network classifiers, the Minibatch Stochastic Three Points (MiSTP) method (Boucherouite et al., 2024) demonstrates superior performance compared to other variants of zero-order methods, that are adapted to the stochastic setting, such as RSGF (also called ZO-SGD) (Ghadimi & Lan, 2013), ZO-SVRG-Ave, and ZO-SVRG (Liu et al., 2018).

Within the realm of first-order optimization methods that rely on gradient information, numerous studies have investigated the almost sure convergence of the Stochastic Gradient Descent (SGD) algorithm and its variants (Bertsekas & Tsitsiklis, 2000; Nguyen et al., 2019; Mertikopoulos et al., 2020; Sebbouh et al., 2021; Liu & Yuan, 2022). In contrast, the literature on the almost sure convergence of zeroth-order methods remains less developed compared to that of SGD.

In (Gratton et al., 2015), the authors investigate zeroth-order direct-search methods under a probabilistic descent framework. Specifically, they generate randomly the search directions, while assuming that with a certain probability at least one of them is of descent type. For smooth objective functions, their analysis establishes (in Theorem 3.4) the almost sure convergence of the best iterate of the gradient norm to zero. However, the analysis does not provide a convergence rate for this almost sure convergence result, nor does it guarantee the convergence of the gradient norm of the last iterate. In our paper, we provide such results for the STP algorithm (see Table 1). (Gratton et al., 2015) also establish (in Corollary 4.7) a convergence rate $O(1/\sqrt{T})$ for the best iterate with overwhelmingly high probability, but this rate is still not guaranteed almost surely. In our work, we provide the first almost sure convergence rate of the best iterate for zeroth-order methods (see Table 1). More recently, Wang & Feng (2022) explore the convergence of the Stochastic Zeroth-order Gradient Descent (SZGD) algorithm for objective functions satisfying the Łojasiewicz inequality. Assuming smoothness, they demonstrated (in Lemma 1) that the gradient norm of the last iterate converges to zero. Furthermore, in Lemma 2, they proved that the sequence generated by the SZGD algorithm converges almost surely to a critical point, which is a stronger result, since the gradient of $f$ is continuous. However, this analysis is limited to Łojasiewicz functions, which, by definition, satisfy a strong property which is the property essentially used in the analysis of strongly convex functions.

In this paper, we are interested in studying the almost sure convergence of the STP algorithm. For the three classes of functions (smooth, smooth convex, and smooth strongly convex), first convergence results, in terms of expectation, were provided in Bergou et al. (2020). However, it is crucial to note that ensuring almost sure convergence properties is essential for understanding the behavior of each trajectory of the STP algorithm and guaranteeing that any instantiation of the algorithm converges with probability one.

**Our Contribution & Related Work.** In cases where the only verified assumptions regarding the function are its smoothness and having a lower bound, Bergou et al. established in their paper (Bergou et al., 2020, Theorem 4.1) that by using Algorithm 1 and selecting a step size sequence $\{\frac{\alpha}{\sqrt{t}}\}_{t \geq 1}$ with $\alpha > 0$, the best gradient iterate converges in expectation to 0 at a rate of $O(\frac{\sqrt{d}}{\sqrt{T}})$. Expanding on this, we prove that employing a similar step size sequence $\{\frac{\alpha}{t^{\frac{1}{2}+\epsilon}}\}_{t \geq 1}$ with $\epsilon \in (0, \frac{1}{2})$ results in an almost sure convergence rate of $o(\frac{1}{T^{\frac{1}{2}-\epsilon}})$, which is arbitrarily close to the rate achieved for the convergence in expectation when $\epsilon$ is close to 0 (see Theorem 1). It's worth noting that a similar almost sure convergence result has been established for the SGD Algorithm. For more information, refer to (Sebbouh et al., 2021, Corollary 18) and (Liu & Yuan, 2022, Theorem 1). However, it should be noted that this similar result for the SGD Algorithm is provided for $\min_{1 \leq t \leq T} \|\nabla f(\theta^t)\|^2$, while for the STP Algorithm, it is provided for $\min_{1 \leq t \leq T} \|\nabla f(\theta^t)\|$. More precisely, for the STP algorithm, we have $\min_{1 \leq t \leq T} \|\nabla f(\theta^t)\| = o(1/T^{\frac{1}{2}-\epsilon})$, while for the SGD algorithm, we have $\min_{1 \leq t \leq T} \|\nabla f(\theta^t)\| = o(1/T^{\frac{1}{4}-\frac{\epsilon}{2}})$. The issue with both convergence results, whether it's the one by Bergou et al. (Bergou et al., 2020, Theorem 4.1) about the convergence in expectation or our first result about the almost sure convergence, is that they don't guarantee the gradient of $f$ at the final point $\theta^T$ to be small (either in expectation or almost surely). Instead, they assure that the gradient of $f$ at some point produced by the STP algorithm is small. In our paper, we additionally prove that the gradient of $f$ at the final point $\theta^T$ converges to 0 almost surely and in expectation without requiring additional assumptions about the function beyond its smoothness and having a lower bound (see Theorems 2 and 3). Notably, for the case of the SGD algorithm, the question of the almost sure convergence of the last gradient iterate has been addressed in Bertsekas & Tsitsiklis (2000).

For smooth convex functions, if $f$ has a global minimum $\theta^*$ and possesses a bounded sublevel set, we show that selecting a step size sequence $\alpha_t = O(\frac{1}{t^{1-\beta}})$ for some $\beta \in (0, \frac{1}{2})$ ensures that $f(\theta^T)$ converges almost surely to $f(\theta^*)$ at a rate of $o(\frac{1}{T^{1-\epsilon}})$ for all $\epsilon \in (2\beta, 1)$ (see Theorem 5). A similar result, with the same convergence rate and the same criteria for choosing the step size sequence, is established for the stochastic Nesterov's accelerated gradient algorithm by Jun Liu et al. in (Liu & Yuan, 2022, Theorem 3). For the same class of functions and under the same assumptions, Bergou et al. established in (Bergou et al., 2020, Theorem 5.5) that for a fixed precision $\epsilon$ and a sufficiently large number of iterations $T$ on the order of $\frac{1}{\epsilon}$, by selecting a step size sequence $\{\frac{|f(\theta^t+hs_t)-f(\theta^t)|}{Lh}\}_{t \geq 1}$ where $h$ is sufficiently small on the order of $\mathbb{E}[f(\theta^{T-1})] - f(\theta^*)$, one can

get: $\mathbb{E}[f(\theta^T)] - f(\theta^*) \leq \epsilon$. Here, the choice of $h$ depends on the quantity $\mathbb{E}[f(\theta^{T-1})]$ which is not known at the begining. Moreover, the theorem does not guarantee that $\mathbb{E}[f(\theta^T)]$ converges to $f(\theta^*)$, because the step sizes depend on $\epsilon$. In contrast, in Theorem 4, we show that by selecting a step size sequence $\{\frac{\alpha}{t}\}_{t \geq 1}$, where $\alpha$ is suitably chosen, $\mathbb{E}[f(\theta^T)]$ converges to $f(\theta^*)$ at a rate of $O(\frac{d}{T})$ .

For smooth, strongly convex functions, Bergou et al. established in (Bergou et al., 2020, Theorem 6.3) that, for any $\epsilon > 0$, using the step size sequence $\{\frac{|f(\theta^t + hs_t) - f(\theta^t)|}{Lh}\}_{t \geq 1}$, where $h$ is small on the order of $\sqrt{\epsilon}$, the gap between the expected value of the objective function and its infimum stays within $\epsilon$ accuracy for a number of iterations on the order of $\log(\frac{f(\theta^t) - \inf_{\theta \in \mathbb{R}^d} f(\theta)}{\epsilon})$. However, this result doesn't indicate how the gap $\mathbb{E}[f(\theta^T)] - \inf_{\theta \in \mathbb{R}^d} f(\theta)$ improves with more iterations and does not guarantee convergence since the step sizes depend on $\epsilon$. To address this issue, we define the step size sequence as $\{\frac{|f(\theta^t + h^{-t}s_t) - f(\theta^t)|}{Lh^{-t}}\}_{t \geq 1}$ with a suitable $h$, leading to a convergence rate of $O((1 - \frac{\mu}{2\pi dL})^T)$ in expectation, and $o((1 - s\frac{\mu}{2\pi dL})^T)$ almost surely for all $s \in (0, 1)$ (see Theorems 6 and 7). All of our convergence rates are succinctly presented in Table 1.

Table 1: Summary of convergence rates for the STP algorithm.

| Functions | Assump | Step size | Iterate | Conv / Rate | Ref |
|---|---|---|---|---|---|
| Smooth | 1,5,6 | $\{\frac{\alpha}{\sqrt{t}}\}_{t \geq 1}, \alpha > 0$ | $\min_{1 \leq t \leq T} \|\nabla f(\theta^t)\|$ | $\mathbb{E}/O\left(\frac{\sqrt{d}}{\sqrt{T}}\right)$ | Thm 4.1 Bergou et al. (2020) |
| | 1,5,6 | $\begin{cases} \{\frac{\alpha}{t^{\frac{1}{2}+\epsilon}}\}_{t \geq 1}, \alpha > 0 \\ \epsilon \in (0, \frac{1}{2}) \end{cases}$ | $\min_{1 \leq t \leq T} \|\nabla f(\theta^t)\|$ | a.s. $/o(\frac{1}{T^{\frac{1}{2}-\epsilon}})$ | Thm 1 |
| | 1,5,6 | $\begin{cases} \{\alpha_t\}_{t \geq 1} \\ \sum_{t=1}^{\infty} \alpha_t^2 < \infty \\ \sum_{t=1}^{\infty} \alpha_t = \infty \end{cases}$ | $\|\nabla f(\theta^T)\|$ | $\mathbb{E}$ & a.s. $/ o(1)$ | Thm 2 Thm 3 |
| Smooth, convex | 1,2,3,5,6 | $\alpha_t = \frac{\alpha}{t}, \alpha$ is suitably chosen | $f(\theta^T) - f(\theta^*)$ | $\mathbb{E}/O\left(\frac{d}{T}\right)$ | Thm 4 |
| | 1,2,3,5,6 | $\alpha_t = O\left(\frac{1}{t^{1-\beta}}\right), \beta \in (0, \frac{1}{2})$ | $f(\theta^T) - f(\theta^*)$ | a.s. $/ o\left(\frac{1}{T^{1-\epsilon}}\right),$ $\forall \epsilon \in (2\beta, 1)$ | Thm 5 |
| Smooth, strongly convex | 1,4,5,6,7 | $\begin{cases} \{\frac{|f(\theta^t + h^{-t}s_t) - f(\theta^t)|}{Lh^{-t}}\}_{t \geq 1} \\ h \text{ is large enough} \end{cases}$ | $f(\theta^T) - f(\theta^*)$ | $\mathbb{E}/O((1 - \beta)^T)$ $\beta \sim \frac{\mu}{dL}$ | Thm 6 |
| | 1,4,5,6,7 | $\begin{cases} \{\frac{|f(\theta^t + h^{-t}s_t) - f(\theta^t)|}{Lh^{-t}}\}_{t \geq 1} \\ h \text{ is large enough} \end{cases}$ | $f(\theta^T) - f(\theta^*)$ | a.s. $/ o((1 - \beta)^T)$ $\beta \sim \frac{s\mu}{dL}; \forall s \in (0, 1)$ | Thm 7 |

## 2 PROBLEM SETUP AND ASSUMPTIONS

We are interested in the following optimization problem:

$$\min_{\theta \in \mathbb{R}^d} f(\theta),$$

where the objective function $f : \mathbb{R}^d \mapsto \mathbb{R}$ is differentiable and bounded from below. In this context, we work within the constraint of not having access to the derivatives of $f$, relying exclusively on a function evaluation oracle.

Throughout the rest of the paper, we assume that the objective function is differentiable and bounded from below. We consider the following additional assumptions about $f$:

**Assumption 1.** $f$ is $L-smooth$, i.e., $\forall x, y \in \mathbb{R}^d, \|\nabla f(x) - \nabla f(y)\|_2 \leq L\|x - y\|_2$.

Note that Assumption 1, implies the following result (Nesterov, 2013, Lemma 1.2.3):

$$\forall x, y \in \mathbb{R}^d, \ |f(y) - f(x) - \langle \nabla f(x), y - x \rangle| \leq \frac{L}{2}\|y - x\|_2^2. \tag{1}$$

**Assumption 2.** $\exists \theta^* \in \mathbb{R}^d, \ f(\theta^*) = \inf_{\theta \in \mathbb{R}^d} f(\theta)$.

**Assumption 3.** $f$ is convex and there exists $c \in \mathbb{R}^d$ such that the sublevel set of $f$ defined by $c$ is bounded, i.e.,

  1. $\forall x, y \in \mathbb{R}^d, \ f(y) \geq f(x) + \langle \nabla f(x), y - x \rangle.$

2. *There exists $c \in \mathbb{R}^d$ such that $L(c) = \{x \in \mathbb{R}^d \mid f(x) \leq f(c)\}$ is bounded.*

**Assumption 4.** *$f$ is $\mu$-strongly convex, i.e., there exists a positive constant $\mu$ such that:*

$$\forall x, y \in \mathbb{R}^d, \ f(y) \geq f(x) + \langle \nabla f(x), y - x \rangle + \frac{\mu}{2} \|y - x\|_2^2.$$

Note that Assumption 4, implies the following result (Nesterov, 2013, Theorem 2.1.8):

$$\forall x \in \mathbb{R}^d, \ \frac{1}{2\mu} \|\nabla f(x)\|_2^2 \geq f(x) - \inf_{y \in \mathbb{R}^d} f(y) \quad \text{(Polyak-Łojasiewicz inequality)}.$$

For the distributions $\mathcal{D}$ over $\mathbb{R}^d$, we make the following assumptions:

**Assumption 5.** *The probability distribution $\mathcal{D}$ on $\mathbb{R}^d$ satisfies:*

1. *$\gamma_{\mathcal{D}} := \mathbb{E}_{s \sim \mathcal{D}}[\|s\|_2^2] < \infty$.*

2. *There exists a norm $\|.\|_{\mathcal{D}}$ on $\mathbb{R}^d$, for which we can find a constant $\mu_{\mathcal{D}} > 0$ such that:*

$$\forall v \in \mathbb{R}^d, \ \mathbb{E}_{s \sim \mathcal{D}} |\langle v, s \rangle| \geq \mu_{\mathcal{D}} \|v\|_{\mathcal{D}}.$$

In (Bergou et al., 2020, Lemma 3.4), the validity of Assumption 5 has been established for several distributions including:

(i) For the normal distribution with zero mean and the identity matrix over $d$ as covariance matrix, i.e., $\mathcal{D} \sim N(0, \frac{I_d}{d})$:

$$\begin{cases} \gamma_{\mathcal{D}} = 1, \\ \mathbb{E}_{s \sim \mathcal{D}} |\langle v, s \rangle| = \frac{\sqrt{2}}{\sqrt{d\pi}} \|v\|_2. \end{cases}$$

(ii) For the uniform distribution on the unit sphere in $\mathbb{R}^d$:

$$\begin{cases} \gamma_{\mathcal{D}} = 1, \\ \mathbb{E}_{s \sim \mathcal{D}} |\langle g, s \rangle| \sim \frac{1}{\sqrt{2\pi d}} \|g\|_2. \end{cases}$$

**Assumption 6.** *For all $s \sim \mathcal{D} : \mathbb{P}(\|s\|_2 \leq 1) = 1$.*

Note that under Assumption 6, we have $\gamma_{\mathcal{D}} \leq 1$. Finally, we add the following assumption regarding $\mu_{\mathcal{D}}$ involved in Assumption 5:

**Assumption 7.** *$\mu_{\mathcal{D}} < 1$.*

**Remark 1.** *In Section 5, we modify the second condition of Assumption 5 by replacing it with:*

*There exists a constant $\mu_{\mathcal{D}} > 0$ such that: $\forall v \in \mathbb{R}^d, \ \mathbb{E}_{s \sim \mathcal{D}} |\langle v, s \rangle| \geq \mu_{\mathcal{D}} \|v\|_2$.*

Since norms are equivalent on $\mathbb{R}^d$, this condition is equivalent to the second condition of Assumption 5. We note also that Assumption 7 is satisfied for distribution distribution (i).

Throughout the paper, the abbreviation "a.s" stands for "almost surely".

## 3 CONVERGENCE ANALYSIS FOR THE CLASS OF SMOOTH FUNCTIONS

### 3.1 CONVERGENCE ANALYSIS FOR THE BEST ITERATE

In this subsection, we will assume that Assumptions 1, 5 and 6 hold true. Under these assumptions, we establish that for any $\epsilon > 0$, when $\{\theta^t\}_{t \geq 1}$ is generated by the STP algorithm using the step size sequence $\{\frac{\alpha}{t^{\frac{1}{2}+\epsilon}}\}_{t \geq 1}$ with $\alpha > 0$, it follows that $\min_{1 \leq t \leq T} \|\nabla f(\theta^t)\|$ converges almost surely to 0 at a rate of $o(\frac{1}{T^{\frac{1}{2}-\epsilon}})$. This result is provided by Theorem 1, which follows from the first finding of Lemma 1 that ensures that: $\sum_{t=1}^{\infty} \frac{1}{t^{\frac{1}{2}+\epsilon}} \mathbb{E}[\|\nabla f(\theta^t)\|_{\mathcal{D}}] < \infty$.

**Lemma 1.** *Assume that Assumptions 1, 5 and 6 hold true. Let $\{\alpha_t\}_{t\geq 1}$ be a sequence of step sizes satisfying $\sum_{t=1}^{\infty} \alpha_t^2 < \infty$. Let $\{\theta^t\}_{t\geq 1}$ be a sequence generated by Algorithm 1. Then, the following results hold:*

$$\begin{cases} \sum_{t=1}^{\infty} \alpha_t \mathbb{E}\left[\|\nabla f(\theta^t)\|_{\mathcal{D}}\right] < \infty, \\ \sum_{t=1}^{\infty} \alpha_t \|\nabla f(\theta^t)\|_{\mathcal{D}} < \infty \quad a.s. \end{cases}$$

In the Appendix (Lemma 5), we prove that if $\{X_t\}_{t\geq 1}$ is a sequence of nonnegative real numbers that is non-increasing and converges to 0, and $\{\alpha_t\}_{t\geq 1}$ is a sequence of real numbers such that $\sum_{t=1}^{\infty} \alpha_t X_t$ converges, then $X_T$ converges to 0 at a rate of $o\left(1/\sum_{t=1}^{T} \alpha_t\right)$. As a result, since $\{\min_{t\leq T} \|\nabla f(\theta^t)\|_{\mathcal{D}}\}_{T\geq 1}$ satisfies the conditions of this lemma when $\sum_{t=1}^{\infty} \alpha_t^2 < \infty$ and $\sum_{t=1}^{\infty} \alpha_t = \infty$, we conclude that in this case, the best gradient iterate converges to 0 at a rate of $o\left(1/\sum_{t=1}^{T} \alpha_t\right)$. This result is formally presented in Theorem 1.

**Theorem 1.** *Assume that Assumptions 1, 5 and 6 hold. Let $\{\theta^t\}_{t\geq 1}$ be a sequence generated by Algorithm 1, where the step size sequence $\{\alpha_t\}_{t\geq 1}$ satisfies the following conditions:*

$$\begin{cases} \sum_{t=1}^{\infty} \alpha_t^2 < \infty, \\ \sum_{t=1}^{\infty} \alpha_t = \infty. \end{cases}$$

*Then, we have:*

$$\min_{1\leq t\leq T} \|\nabla f(\theta^t)\|_{\mathcal{D}} = o\left(\frac{1}{\sum_{t=1}^{T} \alpha_t}\right) \quad a.s.$$

*In particular, if we choose $\alpha_t = \frac{\alpha}{t^{\frac{1}{2}+\epsilon}}$ with $\alpha > 0$ and $\epsilon \in \left(0, \frac{1}{2}\right)$, it follows that:*

$$\min_{1\leq t\leq T} \|\nabla f(\theta^t)\|_{\mathcal{D}} = o\left(\frac{1}{T^{\frac{1}{2}-\epsilon}}\right) \quad a.s.$$

In (Bergou et al., 2020, Theorem 4.1), the authors established that by using the STP algorithm with a step size sequence $\left\{\frac{\alpha}{\sqrt{t}}\right\}_{t\geq 1}$, where $\alpha > 0$, the best gradient iterate converges to 0 in expectation at a rate of $O\left(\frac{\sqrt{d}}{\sqrt{T}}\right)$. The second result of Theorem 1 provides a similar version of this result almost surely, where both the step sizes and convergence rates are roughly similar.

**Remark 2.** *Since all norms are equivalent in finite dimension, for any norm $\|\cdot\|$ on $\mathbb{R}^d$, we can conclude that by selecting $\alpha_t = \frac{\alpha}{t^{\frac{1}{2}+\epsilon}}$, where $\alpha > 0$ and $\epsilon \in \left(0, \frac{1}{2}\right)$, the following holds:*

$$\min_{1\leq t\leq T} \|\nabla f(\theta^t)\| = o\left(\frac{1}{T^{\frac{1}{2}-\epsilon}}\right) \quad a.s.$$

**Remark 3.** *In the non-convex setting, the convergence analysis in the previous theorem implies that $\min_{1\leq t\leq T} \|\nabla f(\theta^t)\|$ converges to zero almost surely. However, it remains uncertain whether the gradient of the last iterate $\|\nabla f(\theta^T)\|$ also converges almost surely to 0. In section 3.2, we will establish the convergence of the last iterate of the gradient, both almost surely and in expectation.*

## 3.2 CONVERGENCE ANALYSIS FOR THE FINAL ITERATE

In this subsection, we will assume that Assumptions 1, 5 and 6 hold true. Under these assumptions, we establish that the STP algorithm ensures the almost sure convergence of $\|\nabla f(\theta^T)\|$ to 0 and the convergence of $\mathbb{E}[\|\nabla f(\theta^T)\|]$ to 0. This result holds for any step size sequence $\{\alpha_t\}_{t\geq 1}$ such that: $\sum_{t=1}^{\infty} \alpha_t^2 < \infty$ and $\sum_{t=1}^{\infty} \alpha_t = \infty$. The almost sure convergence result is provided by Theorem 2, while the convergence in expectation is established by Theorem 3. Notably, both of these theorems are derived from Lemma 1 and Lemma 7. (see the Appendix).

**Theorem 2.** *Assume that Assumptions 1, 5 and 6 hold true. Suppose that the step size sequence satisfies:*

$$\begin{cases} \sum_{t=1}^{\infty} \alpha_t^2 < \infty, \\ \sum_{t=1}^{\infty} \alpha_t = \infty. \end{cases}$$

*Let $\{\theta^t\}_{t\geq 1}$ be a sequence generated by Algorithm 1. Then, we have:*

$$\lim_{T\to\infty} \|\nabla f(\theta^T)\|_{\mathcal{D}} = 0 \quad a.s.$$

**Theorem 3.** *Assume that Assumptions 1, 5 and 6 hold true. Suppose that the step size sequence satisfies:*

$$\begin{cases} \sum_{t=1}^{\infty} \alpha_t^2 < \infty, \\ \sum_{t=1}^{\infty} \alpha_t = \infty. \end{cases}$$

*Let $\{\theta^t\}_{t\geq 1}$ be a sequence generated by Algorithm 1. Then, we have:*

$$\lim_{T\to+\infty} \mathbb{E}[\|\nabla f(\theta^T)\|_{\mathcal{D}}] = 0.$$

**Remark 4.** *In particular, for any $\epsilon \in (0, \frac{1}{2})$, the step size sequence $\{\frac{\alpha}{t^{\frac{1}{2}+\epsilon}}\}_{t\geq 1}$ with $\alpha > 0$, satisfies the conditions on step sizes of Theorems 2 and 3.*

# 4 CONVERGENCE ANALYSIS FOR THE CLASS OF SMOOTH CONVEX FUNCTIONS

In this section we will assume that Assumptions 1 to 3, 5 and 6 hold true. Since $f$ is a real-valued, continuous, and convex function, it follows that $f$ is a closed proper convex function. Additionally, Assumption 3 guarantees the existence of a vector $c$ such that the sublevel set of $f$ defined by $c$ is bounded. Therefore, we can deduce that all sublevel sets of $f$ are bounded, as shown in (Rockafellar, 2015, Corollary 8.7.1). Let $\theta^1$ be the initial vector of the STP algorithm. In particular, the sublevel set $L(\theta^1)$ is bounded, and it forms a compact set of $\mathbb{R}^d$ (because $f$ is continuous).

Let's denote $\|\cdot\|_{\mathcal{D}}^*$ as the dual norm of $\|\cdot\|_{\mathcal{D}}$, defined for all $\theta \in \mathbb{R}^d$ by: $\|\theta\|_{\mathcal{D}}^* = \sup_{v\in\mathbb{R}^d\setminus\{0\}} \frac{\langle v,\theta\rangle}{\|v\|_{\mathcal{D}}}$. Since $\theta \mapsto \|\theta - \theta^*\|_{\mathcal{D}}^*$ is continuous over $\mathbb{R}^d$ and $L(\theta^1)$ is a compact subset of $\mathbb{R}^d$, we have:

$$R := \sup_{\theta\in L(\theta^1)} \|\theta - \theta^*\|_{\mathcal{D}}^* < \infty.$$

Since $f$ is convex, we have that for all $\theta \in L(\theta^1)$:

$$f(\theta) - f(\theta^*) \leq \langle \nabla f(\theta), \theta - \theta^*\rangle \leq \|\nabla f(\theta)\|_{\mathcal{D}} \underbrace{\sup_{v\in\mathbb{R}^d\setminus\{0\}} \frac{\langle v, \theta - \theta^*\rangle}{\|v\|_{\mathcal{D}}}}_{\|\theta-\theta^*\|_{\mathcal{D}}^*} \leq R\|\nabla f(\theta)\|_{\mathcal{D}}.$$

By the construction of the STP algorithm, for all $t \geq 1$, $f(\theta^t) \leq f(\theta^1)$. Therefore, for all $t \geq 1$, $\theta^t \in L(\theta^1)$, and thus we have:

$$\forall t \geq 1, \ f(\theta^t) - f(\theta^*) \leq R\|\nabla f(\theta^t)\|_{\mathcal{D}}. \tag{2}$$

This final result serves as a crucial point for the convergence analysis of Theorem 4 and Theorem 5.

The following Theorems 4 and 5, show the convergence of the final iterate $f(\theta^T)$ to the optimal value with a rate $O(d/T)$ in expectation, and a rate approximately $o(1/T)$ almost surely.

**Theorem 4.** *Assume that Assumptions 1 to 3, 5 and 6 hold true, and consider a sequence $\{\theta^t\}_{t\geq 1}$ generated by Algorithm 1, where the step size sequence is defined as $\{\frac{\alpha}{t}\}_{t\geq 1}$ with $\alpha > \frac{R}{\mu_{\mathcal{D}}}$.*

*We have the following bound:*

$$\mathbb{E}\left[f(\theta^T)\right] - f(\theta^*) \leq \frac{a}{T},$$

*where*

$$a = \max\left(\frac{3\alpha\mu_{\mathcal{D}}}{R}\left(f(\theta^1) - f(\theta^*)\right), \frac{L\alpha^2}{2\left(\frac{\alpha\mu_{\mathcal{D}}}{R} - 1\right)}\right).$$

*In particular, if $\mu_{\mathcal{D}}$ is proportional to $\frac{1}{\sqrt{d}}$, then by taking $\alpha = \frac{2R}{\mu_{\mathcal{D}}}$, we obtain a complexity bound of the form $O\left(\frac{d}{T}\right)$.*

Note that for the normal distribution (i) with zero mean and identity covariance matrix , as well as for the uniform distribution over the unit sphere (ii), $\mu_{\mathcal{D}}$ is proportional to $\frac{1}{\sqrt{d}}$.

**Remark 5.** *Assume that $\mu_{\mathcal{D}}$ is proportional to $\frac{1}{\sqrt{d}}$, and let $c \geq 2$ be a constant. If we choose $\alpha$ such that $\frac{2R}{\mu_{\mathcal{D}}} \leq \alpha \leq \frac{cR}{\mu_{\mathcal{D}}}$, we get $\frac{L\alpha^2}{2(\frac{\alpha\mu_{\mathcal{D}}}{R}-1)} = O(cd)$. Thus, we obtain the convergence rate $O(\frac{cd}{T})$ in Theorem 4.*

**Theorem 5.** *Assume that Assumptions 1 to 3, 5 and 6 hold true. Let $\{\theta^t\}_{t \geq 1}$ be a sequence generated by Algorithm 1, where the step size sequence is given by $\alpha_t = O\left(\frac{1}{t^{1-\beta}}\right)$ for some $\beta \in \left(0, \frac{1}{2}\right)$.*

*We then have the following:*

$$\forall \epsilon \in (2\beta, 1), \ f(\theta^T) - f(\theta^*) = o\left(\frac{1}{T^{1-\epsilon}}\right) \quad a.s.$$

## 5 STRONGLY CONVEX ALMOST SURE CONVERGENCE RATE FOR STP ALGORITHM

In this section we will assume that Assumptions 1 and 4 to 7 hold true. The main results of this section are stated in Theorems 6 and 7, which follow from Lemma 3. When step sizes are obtained by approximating the directional derivatives of the function with respect to the random search directions, we show in Theorem 6 that $f(\theta^T)$ converges in expectation to $\inf_{\theta \in \mathbb{R}^d} f(\theta)$ at a rate of $O((1 - \frac{\mu}{2\pi dL})^T)$, and in Theorem 7, we establish this convergence almost surely at a rate arbitrarily close to $o((1 - \frac{\mu}{2\pi dL})^T)$, where $\mu$ and $L$ are the strong convexity and smoothness parameters of the function, and $d$ is the dimension of the space. We recall that a strongly convex function has a unique minimizer, which we denote by $\theta^*$.

The following Lemma 2, controls the decrease per iteration of the value function. It is used to control the total decrease of the value function after $T$ iterations given in Lemma 3.

**Lemma 2.** *Assume that Assumptions 1 and 6 hold true. Let $h \in (1, \infty)$ and let $\{\theta^t\}_{t \geq 1}$ be a sequence generated by Algorithm 1, where the step size sequence used is $\{\frac{|f(\theta^t + h^{-t}s_t) - f(\theta^t)|}{Lh^{-t}}\}_{t \geq 1}$. Then we have:*

$$\forall t \geq 1, \ f(\theta^{t+1}) \leq f(\theta^t) - \frac{|\langle \nabla f(\theta^t), s_t \rangle|^2}{2L} + \frac{L}{8}h^{-2t} \ a.s.$$

**Lemma 3.** *Assume that Assumptions 1 and 4 to 7 hold true. Let $h \in (1, \infty)$ and let $\{\theta^t\}_{t \geq 1}$ be a sequence generated by Algorithm 1, where the step size sequence used is $\{\frac{|f(\theta^t + h^{-t}s_t) - f(\theta^t)|}{Lh^{-t}}\}_{t \geq 1}$. Then we have:*

$$\forall T \geq 2, \ \mathbb{E}[f(\theta^T) - f(\theta^*)] \leq \left(1 - \frac{\mu_{\mathcal{D}}^2\mu}{L}\right)^{T-1}[f(\theta^1) - f(\theta^*)] + \frac{L}{8}\sum_{i=1}^{T-1}\left(1 - \frac{\mu_{\mathcal{D}}^2\mu}{L}\right)^{T-1-i}h^{-2i}.$$

**Theorem 6.** *Assume that Assumptions 1 and 4 to 7 hold true. Let $h \in \left(\frac{1}{\sqrt{1 - \frac{\mu_{\mathcal{D}}^2\mu}{L}}}, \infty\right)$ and let $\{\theta^t\}_{t \geq 1}$ be a sequence generated by Algorithm 1, where the step size sequence used is $\{\frac{|f(\theta^t + h^{-t}s_t) - f(\theta^t)|}{Lh^{-t}}\}_{t \geq 1}$. Then we have:*

$$\forall T \geq 2, \ \mathbb{E}[f(\theta^T) - f(\theta^*)] \leq \left(1 - \frac{\mu_{\mathcal{D}}^2\mu}{L}\right)^{T-1}\left[f(\theta^1) - f(\theta^*) + \frac{L}{8}\frac{1}{h^2\left(1 - \frac{\mu_{\mathcal{D}}^2\mu}{L}\right) - 1}\right]. \quad (3)$$

*In particular, if $\mu_{\mathcal{D}}$ is proportional to $\frac{1}{\sqrt{d}}$, i.e., $\mu_{\mathcal{D}} = \frac{K}{\sqrt{d}}$, for some positive constant $K$, then by taking $h = \frac{2}{\sqrt{1 - \frac{\mu_{\mathcal{D}}^2\mu}{L}}}$, we obtain a rate of $O\left(\left(1 - \frac{\mu K^2}{dL}\right)^T\right)$.*

**Theorem 7.** *Assume that Assumptions 1 and 4 to 7 hold true. Let $\{\theta^t\}_{t\geq 1}$ be a sequence generated by Algorithm 1, where the step size sequence used is $\{\frac{|f(\theta^t+h^{-t}s_t)-f(\theta^t)|}{Lh^{-t}}\}_{t\geq 1}$, with*

$$h \in \left(\frac{1}{\sqrt{1-\frac{\mu_\mathcal{D}^2\mu}{L}}}, \infty\right), \text{ we have: } \forall s \in (0,1), \ f(\theta^T) - f(\theta^*) = o\left(\left(1-s\frac{\mu_\mathcal{D}^2\mu}{L}\right)^T\right) \ a.s.$$

*In particular, if $\mu_\mathcal{D}$ is proportional to $\frac{1}{\sqrt{d}}$, i.e., $\mu_\mathcal{D} = \frac{K}{\sqrt{d}}$, for some positive constant $K$, then for all $s \in (0,1)$, we obtain a convergence rate of $o((1-s\frac{\mu K^2}{dL})^T)$.*

**Remark 6.** *Note that for the uniform distribution over the unit sphere, we have $\mu_\mathcal{D} = \frac{1}{\sqrt{2\pi d}}$. The convergence rates for Theorem 6 and Theorem 7 in this case are respectively $O((1-\frac{\mu}{2\pi dL})^T)$ and $o((1-s\frac{\mu}{2\pi dL})^T)$ for any $s \in (0,1)$.*

## 6 NUMERICAL EXPERIMENTS

Let's consider the following optimization problem:

$$\min_{\theta \in \mathbb{R}^d} f(\theta) = \frac{1}{2}(\theta_1)^2 + \frac{1}{2}\sum_{i=1}^{d-1}(\theta_{i+1}-\theta_i)^2 + \frac{1}{2}(\theta_d)^2 - \theta_1, \quad \text{initial vector: } \theta^1 = 0,$$

where $d = 500$. This objective function was used in Section 2.1 of Nesterov et al. (2018) to prove the lower complexity bound for gradient methods applied to smooth functions. By running multiple trajectories for the three algorithms: the STP algorithm, the RGF algorithm (Nesterov & Spokoiny, 2017), and the GLD algorithm (Golovin et al., 2020), the objective is to simulate the convergence of the last gradient iterate for each trajectory and also illustrate the rate of convergence of the best gradient iterate.

**RGF Algorithm:** This algorithm starts with an initial vector $\theta^1$ and iteratively updates it according to the following rule $\theta^{t+1} = \theta^t - h_t \frac{f(\theta^t+\mu_t u_t)-f(\theta^t)}{\mu_t}u_t$, where $u_t$ is a normally distributed gaussian vector. In this implementation, we set $\mu_t = 10^{-4}$ and $h_t = \frac{1}{L}$, where $L \leq 4$ represents the smoothness parameter of the objective function.

**GLD algorithm:** This algorithm proceeds as follows: it starts with an initial point $\theta^1$, a sampling distribution $\mathcal{D}$, and a search radius that shrinks from a maximum value $R$ to a minimum value $r$. The number of radius levels is determined by $K = \lceil\log_2\left(\frac{R}{r}\right)\rceil$. For each iteration $t$, the algorithm performs ball sampling trials, where it samples search directions $v^k$ from progressively smaller radii $r_k = 2^{-k}R$, $0 \leq k \leq K$, and then updates the current point by selecting the $v^k$ that results in the minimum value of the objective function. The update step is given by: $\theta^{t+1} = \arg\min_{y\in\{\theta^t,\theta^t+v^0,\cdots,\theta^t+v^K\}} f(y)$. For this algorithm, we use the standard Gaussian distribution $\mathcal{D}$ and set $r = 10^{-5}$ and $R = 10^{-4}$.

For the STP algorithm, we set the step sizes to be $\alpha_t = \frac{4}{t^{0.51}}$, and the random search directions $s_t$ are generated uniformly on the unit sphere of $\mathbb{R}^d$. The chosen step sizes adhere to the form provided in the second result of Theorem 1, where $\epsilon = 0.01$. In our experiment, we run 50 trajectories for each of the three algorithms, all starting from the same initial point 0. We simulate $\log_{10}(\|\nabla f(\theta^T)\|_2)$ as a function of the number of iterations, as well as the elapsed time in seconds. Additionally, to verify the rate assured by Theorem 1 for the STP algorithm, we simulate $T^{0.49}\min_{t\leq T}\|\nabla f(\theta^t)\|_2$ as a function of the number of iterations.

Figure 1 and Figure 2 illustrate the logarithmic decay of the gradient norm with respect to both iterations and elapsed time, highlighting its convergence to zero across all trajectories for the three algorithms. Notably, STP and RGF demonstrate competitive performance, with STP being slightly better, in terms of the number of iterations and the time required to achieve a given accuracy, outperforming the GLD method in both metrics. This similarity between the performance of STP and RGF reflects their similar theoretical complexity bounds. It is important to note also that at each iteration, the STP and RGF methods require two function evaluations, while the GLD method requires $\lceil\log_2\left(\frac{R}{r}\right)\rceil$ function evaluations.

In Figure 3, we observe the convergence of the best gradient iterate to $0$ at a rate of $o(\frac{1}{T^{0.49}})$ across all trajectories for the three algorithms. In particular, this illustrates the rate obtained for the STP algorithm.

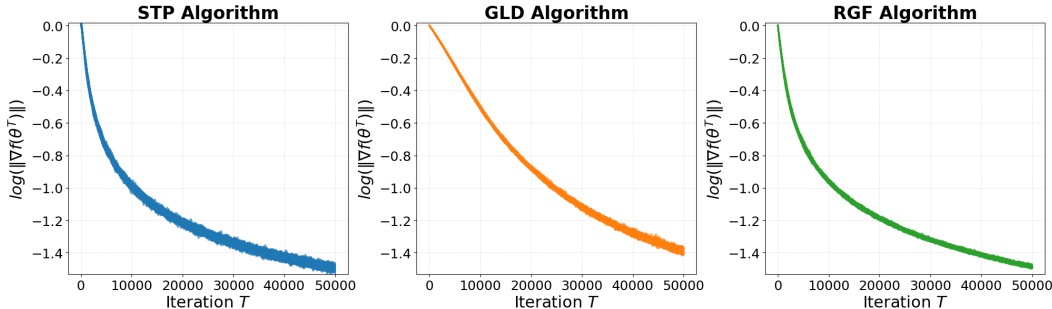

Figure 1: Logarithmic decay of gradient norm vs. Iterations.

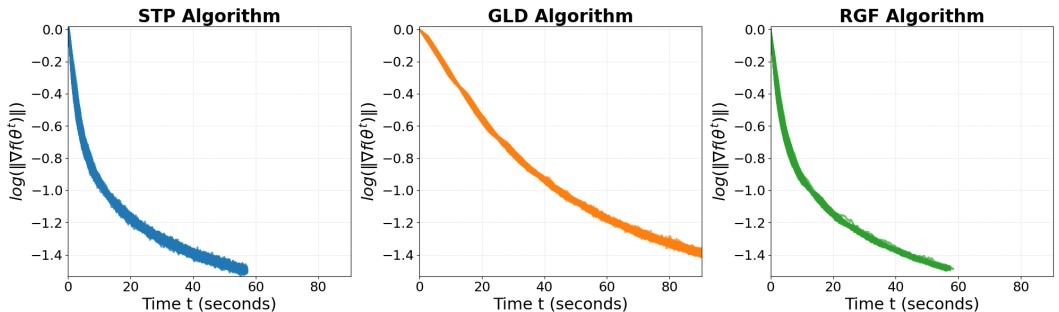

Figure 2: Logarithmic Decay of gradient norm vs. Time.

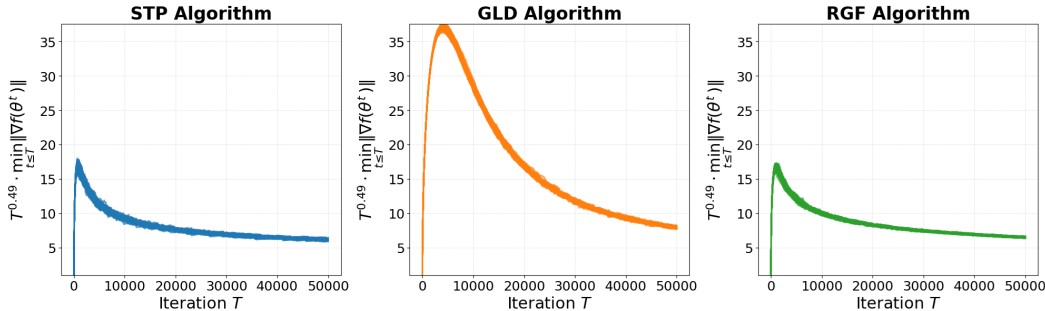

Figure 3: Convergence rate of the best gradient iterate.

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

## A   APPENDIX

**Lemma 4.** *([Bergou et al., 2020](#), Lemma 3.5) Assume that Assumptions 1, 5 and 6 hold true and let $\{\theta^t\}_{t\geq 1}$ be a sequence generated by Algorithm 1. We have:*

$$\mathbb{E}[f(\theta^{t+1}) \mid \theta^t] \leq f(\theta^t) - \mu_{\mathcal{D}}\alpha_t\|\nabla f(\theta^t)\|_{\mathcal{D}} + \frac{L\alpha_t^2}{2}.$$

*Proof of Lemma 1.* Let $t \geq 1$. By Lemma 4, we have that:

$$\mathbb{E}[f(\theta^{t+1}) \mid \theta^t] \leq f(\theta^t) - \alpha_t\mu_{\mathcal{D}}\|\nabla f(\theta^t)\|_{\mathcal{D}} + \frac{L\alpha_t^2}{2}.$$

By taking the expectation, we get: $\mathbb{E}[f(\theta^{t+1})] \leq \mathbb{E}[f(\theta^t)] - \alpha_t\mu_{\mathcal{D}}\mathbb{E}[\|\nabla f(\theta^t)\|_{\mathcal{D}}] + \frac{L\alpha_t^2}{2}$.

It follows that:

$$\mu_{\mathcal{D}}\alpha_t\mathbb{E}[\|\nabla f(\theta^t)\|_{\mathcal{D}}] \leq \mathbb{E}[f(\theta^t)] - \mathbb{E}[f(\theta^{t+1})] + \frac{L\alpha_t^2}{2}. \tag{4}$$

By construction of the algorithm the sequence $\{f(\theta^t)\}_{t\geq 1}$ is non-increasing, and since we assume that $f$ is bounded from below, we have that $\{\mathbb{E}[f(\theta^t)]\}_{t\geq 1}$ is non-increasing and bounded from bellow, and thus converges. As a result, we have: $\sum_{t=1}^{\infty}\mathbb{E}[f(\theta^t)] - \mathbb{E}[f(\theta^{t+1})] < \infty$. Knowing that $\sum_{t=1}^{\infty}\alpha_t^2 < \infty$, we conclude from equation 4, that

$$\sum_{t=1}^{\infty}\alpha_t\mathbb{E}\left[\|\nabla f(\theta^t)\|_{\mathcal{D}}\right] < \infty.$$

We deduce also that $\mathbb{E}\left[\sum_{t=1}^{\infty}\alpha_t\|\nabla f(\theta^t)\|_{\mathcal{D}}\right] = \sum_{t=1}^{\infty}\alpha_t\mathbb{E}\left[\|\nabla f(\theta^t)\|_{\mathcal{D}}\right] < \infty$, which implies that:

$$\sum_{t=1}^{\infty}\alpha_t\|\nabla f(\theta^t)\|_{\mathcal{D}} < \infty \text{ a.s.}$$

$\square$

**Lemma 5.** *Let $\{X_t\}_{t \geq 1}$ be a sequence of nonnegative real numbers that is non increasing and converges to 0, and let $\{\alpha_t\}_{t \geq 1}$ be a sequence of real numbers such that $\sum_{t=1}^{\infty} \alpha_t X_t$ converges. Then, we have:*

$$X_T = o\left(\frac{1}{\sum_{t=1}^{T} \alpha_t}\right).$$

*Proof.* For all $T \geq 1$, we define $U_T = X_T \sum_{i=1}^{T} \alpha_i$ and $R_T = \sum_{i=T}^{\infty} \alpha_i X_i$. We then have:

$$U_T = X_T \sum_{i=1}^{T} (R_i - R_{i+1}) \frac{1}{X_i}.$$

Let $T \geq 2$. We have:

$$
\begin{aligned}
U_T &= X_T \left[ \sum_{i=1}^{T} R_i \frac{1}{X_i} - \sum_{i=1}^{T} R_{i+1} \frac{1}{X_i} \right] \\
&= X_T \left[ \sum_{i=1}^{T} R_i \frac{1}{X_i} - \sum_{i=2}^{T+1} R_i \frac{1}{X_{i-1}} \right] \\
&= X_T \left[ R_1 \frac{1}{X_1} - \frac{R_{T+1}}{X_T} + \sum_{i=2}^{T} R_i \left( \frac{1}{X_i} - \frac{1}{X_{i-1}} \right) \right] \\
&= R_1 \frac{X_T}{X_1} - R_{T+1} + X_T \sum_{i=2}^{T} R_i \left( \frac{1}{X_i} - \frac{1}{X_{i-1}} \right).
\end{aligned}
$$

To prove $\lim_{T \to +\infty} U_T = 0$, it suffices to show that:

$$\lim_{T \to +\infty} X_T \sum_{i=2}^{T} R_i \left( \frac{1}{X_i} - \frac{1}{X_{i-1}} \right) = 0.$$

Let $\epsilon > 0$ and $T_0 \geq 2$ such that for all $T \geq T_0$, we have $R_T \leq \frac{\epsilon}{2}$. Let $T > T_0$, we have:

$$
\begin{aligned}
\left| X_T \sum_{i=2}^{T} R_i \left( \frac{1}{X_i} - \frac{1}{X_{i-1}} \right) \right| &\leq X_T \sum_{i=2}^{T_0} |R_i| \left( \frac{1}{X_i} - \frac{1}{X_{i-1}} \right) + X_T \sum_{i=T_0+1}^{T} \frac{\epsilon}{2} \left( \frac{1}{X_i} - \frac{1}{X_{i-1}} \right) \\
&= X_T \sum_{i=2}^{T_0} |R_i| \left( \frac{1}{X_i} - \frac{1}{X_{i-1}} \right) + \frac{\epsilon X_T}{2} \left( \frac{1}{X_T} - \frac{1}{X_{T_0}} \right) \\
&= X_T \sum_{i=2}^{T_0} |R_i| \left( \frac{1}{X_i} - \frac{1}{X_{i-1}} \right) + \frac{\epsilon}{2} \left( 1 - \frac{X_T}{X_{T_0}} \right) \\
&\leq X_T \sum_{i=2}^{T_0} |R_i| \left( \frac{1}{X_i} - \frac{1}{X_{i-1}} \right) + \frac{\epsilon}{2}.
\end{aligned}
$$

As $\lim_{T \to +\infty} X_T = 0$, there exists $T_1 \geq T_0$ such that for all $T \geq T_1$,

$$\left| X_T \sum_{i=2}^{T} R_i \left( \frac{1}{X_i} - \frac{1}{X_{i-1}} \right) \right| \leq X_T \sum_{i=2}^{T_0} |R_i| \left( \frac{1}{X_i} - \frac{1}{X_{i-1}} \right) + \frac{\epsilon}{2} \leq \epsilon.$$

Therefore $\lim_{T \to +\infty} X_T \sum_{i=2}^{T} R_i \left( \frac{1}{X_i} - \frac{1}{X_{i-1}} \right) = 0$, and we deduce that:

$$X_T = o\left( \frac{1}{\sum_{t=1}^{T} \alpha_t} \right) \quad \text{as } T \to +\infty.$$

$\square$

The following lemma, which is a classical result about Riemann series, will be needed in the proof of Theorem 1.

**Lemma 6.** *For all $\alpha \in (0, 1)$, we have:* $\sum_{t=1}^{T} \frac{1}{t^{\alpha}} \sim \frac{T^{1-\alpha}}{1-\alpha}$.

*Proof of Theorem 1.* Let us define $X_T = \min_{t \leq T} \|\nabla f(\theta_t)\|_{\mathcal{D}}$ for all $T \geq 1$. Since $\sum_{t=1}^{\infty} \alpha_t^2 < \infty$, according to Lemma 1, we deduce that $\sum_{t=1}^{\infty} \alpha_t X_t < \infty$ a.s. It is clear that $\{X_T\}_{T \geq 1}$ is a sequence of nonnegative real numbers that is non increasing, then, by proving $\lim_{T \to +\infty} X_T = 0$ a.s., using lemma 5, we can deduce that:

$$X_T = o\left(\frac{1}{\sum_{t=1}^{T} \alpha_t}\right) \text{ a.s.}$$

Now, we prove that $\lim_{T \to \infty} X_T = 0$ a.s. According to Lemma 1, we have $\sum_{t=1}^{\infty} \alpha_t \|\nabla f(\theta^t)\|_{\mathcal{D}} < \infty$ a.s. Thus, it follows that:

$$\{(\min_{t \leq T} \|\nabla f(\theta^t)\|) \sum_{t=1}^{T} \alpha_t\} \quad \text{is bounded almost surely.}$$

Since $\lim_{T \to +\infty} \sum_{t=1}^{T} \alpha_t = +\infty$, we can conclude that:

$$\lim_{T \to +\infty} X_T = \lim_{T \to +\infty} \min_{t \leq T} \|\nabla f(\theta^t)\|_{\mathcal{D}} = 0 \text{ a.s.}$$

Therefore, we establish the first result of the theorem. The second result is obtained by choosing $\{\alpha_t\}_{t \geq 1}$ defined by $\alpha_t = \frac{1}{t^{\frac{1}{2}+\epsilon}}$. In this case, we have $\sum_{t=1}^{\infty} \alpha_t = \infty$, while $\sum_{t=1}^{\infty} \alpha_t^2 < \infty$.

Using Lemma 6, we have

$$\sum_{t=1}^{T} \frac{1}{t^{\frac{1}{2}+\epsilon}} \sim \frac{T^{\frac{1}{2}-\epsilon}}{\frac{1}{2}-\epsilon}.$$

Therefore,

$$\min_{1 \leq t \leq T} \|\nabla f(\theta^t)\|_{\mathcal{D}} = o\left(\frac{1}{T^{\frac{1}{2}-\epsilon}}\right) \quad \text{a.s.}$$

$\square$

Lemma 7 is first presented in (Alber et al., 1998, Proposition 2) and again in (Mairal, 2013, Lemma A.5), along with a new proof. We provide a new, simpler proof of this lemma that is more straightforward than those presented in these references.

**Lemma 7.** *(Alber et al., 1998, Proposition 2) , (Mairal, 2013, Lemma A.5)* *Let $\{a_t\}_{t \geq 1}, \{b_t\}_{t \geq 1}$ be two nonnegative real sequences. We have:*

$$\begin{cases} \sum_{t=1}^{\infty} a_t b_t < \infty, \\ \sum_{t=1}^{\infty} a_t = \infty, \\ \text{There exists } K \geq 0 \text{ such that } |b_{t+1} - b_t| \leq Ka_t. \end{cases} \implies \lim_{t \to +\infty} b_t = 0.$$

*Proof of Lemma 7.* First, we note that for all $n_0 \geq 1$, we have $\inf_{n \geq n_0} b_n = 0$. Indeed, suppose for contradiction that $\inf_{n \geq n_0} b_n > 0$. In this case, we would have for all $n \geq n_0$, $a_n b_n \geq a_n \inf_{m \geq n_0} b_m$, which implies that the series $\sum a_n b_n$ cannot converge, since $\sum_{n=1}^{\infty} a_n = \infty$ and $inf_{m \geq n_0} b_m > 0$. This contradiction implies that for all $n_0 \geq 1$, we have $\inf_{n \geq n_0} b_n = 0$.

Let $\epsilon > 0$. Let $n_0 \geq 1$ such that for all $n \geq n_0$ we have $\sum_{k=n}^{\infty} a_k b_k \leq \frac{\epsilon^2}{4K}$.

The goal is to prove that for all $n \geq n_0$, $b_n \leq \epsilon$. Let $n \geq n_0$. If $b_n \leq \frac{\epsilon}{2}$, then trivially $b_n \leq \epsilon$. Now assume that $b_n > \frac{\epsilon}{2}$.

We have $\inf_{t \geq n} b_t = 0$, then we can take the smallest index $m > n$ such that $b_m \leq \frac{\epsilon}{2}$. We have:

$$|b_m - b_n| \leq \sum_{i=n}^{m-1} |b_{i+1} - b_i|$$

$$\leq K \sum_{i=n}^{m-1} a_i$$

$$= K \sum_{i \in \{n, \ldots, m-1\}, b_i > \frac{\epsilon}{2}} a_i$$

$$\leq \frac{2K}{\epsilon} \sum_{i \in \{n, \ldots, m-1\}, b_i > \frac{\epsilon}{2}} a_i b_i$$

$$\leq \frac{2K}{\epsilon} \sum_{i=n}^{\infty} a_i b_i$$

$$\leq \frac{\epsilon}{2}.$$

Therefore, by the triangle inequality, we have:

$$b_n \leq b_m + \frac{\epsilon}{2} \leq \epsilon.$$

Thus, for all $n \geq n_0$, we have $b_n \leq \epsilon$, and consequently, we deduce that $\lim_{n \to +\infty} b_n = 0$.

$\square$

*Proof of Theorem 2.* Consider $C > 0$ satisfying: $||.||_{\mathcal{D}} \leq C ||.||_2$.

Let $t \geq 1$. We have that:

$$\left| \left\| \nabla f \left( \theta^{t+1} \right) \right\|_{\mathcal{D}} - \left\| \nabla f \left( \theta^t \right) \right\|_{\mathcal{D}} \right| \leq \| \nabla f(\theta^{t+1}) - \nabla f(\theta^t) \|_{\mathcal{D}}$$

$$\leq CL \| \theta^{t+1} - \theta^t \|_2 \text{ (because } f \text{ is } L\text{-smooth)}$$

$$= CL\alpha_t \| s_t \|_2$$

$$\leq CL\alpha_t \quad \text{a.s. (because we assume Assumption 6 holds true)}$$

Therefore, we have that for all $t \geq 1$: $\mathbb{P} \left( \left| \left\| \nabla f \left( \theta^{t+1} \right) \right\|_{\mathcal{D}} - \left\| \nabla f \left( \theta^t \right) \right\|_{\mathcal{D}} \right| \leq CL\alpha_t \right) = 1$.

Thus: $\mathbb{P} \left( \forall t \geq 1, \left| \left\| \nabla f \left( \theta^{t+1} \right) \right\|_{\mathcal{D}} - \left\| \nabla f \left( \theta^t \right) \right\|_{\mathcal{D}} \right| \leq CL\alpha_t \right) = 1$.

Given that $\begin{cases} \sum_{t=1}^{\infty} \alpha_t \| \nabla f(\theta^t) \|_{\mathcal{D}} < \infty \text{ a.s,} \quad \text{(by Lemma 1 because } \sum_{t=1}^{\infty} \alpha_t^2 < \infty) \\ \sum_{t=1}^{\infty} \alpha_t = \infty. \end{cases}$

Using Lemma 7, with $\{\alpha_t\}_{t \geq 1}$ playing the role of $\{a_t\}_{t \geq 1}$ and $\{\| \nabla f(\theta^t) \|\}_{t \geq 1}$ playing the role of $\{b_t\}_{t \geq 1}$, we conclude that:

$$\lim_{T \to +\infty} \| \nabla f(\theta^T) \|_{\mathcal{D}} = 0 \quad \text{a.s.}$$

$\square$

*Proof of Theorem 3.* Consider $C > 0$ satisfying: $||.||_{\mathcal{D}} \leq C ||.||_2$.

Let $t \geq 1$. We have that:

$$\left| \mathbb{E} \left[ \left\| \nabla f \left( \theta^{t+1} \right) \right\|_{\mathcal{D}} \right] - \mathbb{E} \left[ \left\| \nabla f \left( \theta^t \right) \right\|_{\mathcal{D}} \right] \right| \leq \mathbb{E} \left[ \left| \left\| \nabla f \left( \theta^t \right) \right\|_{\mathcal{D}} - \left\| \nabla f \left( \theta^{t+1} \right) \right\|_{\mathcal{D}} \right| \right]$$

$$\leq \mathbb{E} \left[ \| \nabla f \left( \theta^t \right) - \nabla f \left( \theta^{t+1} \right) \|_{\mathcal{D}} \right]$$

$$\leq CL\mathbb{E} \left[ \| \theta^{t+1} - \theta^t \|_2 \right] \text{ (because } f \text{ is } L\text{-smooth)}$$

$$= CL\alpha_t \mathbb{E} \left[ \| s_t \|_2 \right]$$

$$\leq CL\alpha_t \text{ (because we assume Assumption 6 holds true),}$$

where in the first inequality, we used Jensen's inequality. So, we proved that:

$$\forall t \geq 1, \ \left|\mathbb{E}[\left\|\nabla f\left(\theta^{t+1}\right)\right\|_{\mathcal{D}}] - \mathbb{E}[\left\|\nabla f\left(\theta^{t}\right)\right\|_{\mathcal{D}}]\right| \leq CL\alpha_t.$$

Now, given that $\sum_{t=1}^{\infty} \alpha_t \mathbb{E}[\|\nabla f(\theta^t)\|_{\mathcal{D}}] < \infty$ (by Lemma 1 because $\sum_{t=1}^{\infty} \alpha_t^2 < \infty$), and that $\sum_{t=1}^{\infty} \alpha_t = \infty$, we use Lemma 7, with $\{\alpha_t\}_{t\geq1}$ playing the role of $\{a_t\}_{t\geq1}$ and $\{\mathbb{E}[\|\nabla f(\theta^t)\|]\}_{t\geq1}$ playing the role of $\{b_t\}_{t\geq1}$, to conclude that

$$\lim_{T\to+\infty} \mathbb{E}[\|\nabla f(\theta^T)\|_{\mathcal{D}}] = 0.$$

$\square$

**Lemma 8.** *(Liu & Yuan, 2022, Lemma 1) If $\{Y_t\}_{t\geq1}$ is a sequence of nonnegative random variables adapted to a filtration $\{\mathcal{F}_t\}_{t\geq1}$, and satisfying:*

$$\mathbb{E}\left[Y_{t+1} \mid \mathcal{F}_t\right] \leq (1 - c_1\alpha_t) Y_t + c_2\alpha_t^2 \quad \text{for all} \ \ t \geq 1,$$

*where $\alpha_t = O\left(\frac{1}{t^{1-\beta}}\right)$ for some $\beta \in \left(0, \frac{1}{2}\right)$, and $c_1$ and $c_2$ are positive constants. Then, for any $\epsilon \in (2\beta, 1)$:*

$$Y_t = o\left(\frac{1}{t^{1-\epsilon}}\right) \quad a.s.$$

*Proof of Theorem 4.* By Lemma 4, we have that:

$$\forall t \geq 1, \ \mathbb{E}[f(\theta^{t+1}) \mid \theta^t] \leq f(\theta^t) - \mu_{\mathcal{D}}\alpha_t\|\nabla f(\theta^t)\|_{\mathcal{D}} + \frac{L\alpha_t^2}{2}.$$

Knowing from equation 2 that: $\forall t \geq 1, \ f(\theta^t) - f(\theta^*) \leq R\|\nabla f(\theta^t)\|_{\mathcal{D}}$, we have:

$$\forall t \geq 1, \ \mathbb{E}[f(\theta^{t+1}) - f(\theta^*) \mid \theta^t] \leq \left(1 - \alpha_t \frac{\mu_{\mathcal{D}}}{R}\right) \left(f(\theta^t) - f(\theta^*)\right) + \frac{L\alpha_t^2}{2}.$$

Taking the expectation, and knowing that $\alpha_t = \frac{\alpha}{t}$, we get:

$$\forall t \geq 1, \ \underbrace{\mathbb{E}\left[f(\theta^{t+1}) - f(\theta^*)\right]}_{:=\delta_{t+1}} \leq \left(1 - \frac{\alpha\mu_{\mathcal{D}}}{Rt}\right) \underbrace{\mathbb{E}\left[f(\theta^t) - f(\theta^*)\right]}_{:=\delta_t} + \frac{L\alpha^2}{2t^2}. \tag{5}$$

If $t \in \{1, ..., [\frac{\alpha\mu_{\mathcal{D}}}{R}]+1\}, \delta_t = \mathbb{E}\left[f(\theta^t) - f(\theta^*)\right] \leq f(\theta^1) - f(\theta^*) \leq ([\frac{\alpha\mu_{\mathcal{D}}}{R}]+1)\frac{f(\theta^1)-f(\theta^*)}{t}$. Then

$$\forall t \in \{1, ..., [\frac{\alpha\mu_{\mathcal{D}}}{R}]+1\}, \ \delta_t \leq \frac{3\alpha\mu_{\mathcal{D}}}{R}\frac{f(\theta^1) - f(\theta^*)}{t} \quad \text{, because } [\frac{\alpha\mu_{\mathcal{D}}}{R}]+1 \leq \frac{\alpha\mu_{\mathcal{D}}}{R} + 2 \leq \frac{3\alpha\mu_{\mathcal{D}}}{R}.$$

Let's denote $a$ and $b$ as follows: $a = \max\left(\frac{3\alpha\mu_{\mathcal{D}}}{R}(f(\theta^1) - f(\theta^*)), \frac{L\alpha^2}{2(\frac{\alpha\mu_{\mathcal{D}}}{R}-1)}\right)$ and $b = \frac{\mu_{\mathcal{D}}}{R}$. We have:

$$\forall t \in \{1, ..., [\frac{\alpha\mu_{\mathcal{D}}}{R}]+1\}, \ \delta_t \leq \frac{a}{t}. \tag{6}$$

We will prove by induction that:

$$\forall t \geq [\frac{\alpha\mu_{\mathcal{D}}}{R}]+1, \ \delta_t \leq \frac{a}{t}.$$

For $t = [\frac{\alpha\mu_{\mathcal{D}}}{R}]+1$, we have that:

$$\delta_{[\frac{\alpha\mu_{\mathcal{D}}}{R}]+1} \leq \frac{a}{t}.$$

Let $t \geq [\frac{\alpha\mu_{\mathcal{D}}}{R}]+1$. Assume that $\delta_t \leq \frac{a}{t}$ and let's prove that $\delta_{t+1} \leq \frac{a}{t+1}$. We note that $1 - \frac{\alpha\mu_{\mathcal{D}}}{Rt} > 0$.

From equation 5, we get $\delta_t \leq \frac{a}{t} \implies \delta_{t+1} \leq \frac{a}{t} - \frac{ab\alpha}{t^2} + \frac{L\alpha^2}{2t^2}$. We have also the following equivalence:

$$\frac{a}{t} - \frac{ab\alpha}{t^2} + \frac{L\alpha^2}{2t^2} \leq \frac{a}{t+1} \iff -\frac{ab\alpha}{t^2} + \frac{L\alpha^2}{2t^2} \leq \frac{-a}{t(t+1)}$$

$$\iff -ab\alpha + \frac{L\alpha^2}{2} \leq \frac{-at}{t+1}.$$

Let's prove that the last assertion is true. We have:

$$-a \leq \frac{-at}{t+1} \implies -ab\alpha + a(b\alpha - 1) \leq \frac{-at}{t+1}$$

$$\implies -ab\alpha + \frac{L\alpha^2}{2} \leq \frac{-at}{t+1} .$$

The last implication comes from $a(b\alpha - 1) = \max\left( \underbrace{(b\alpha - 1)}_{>0} \frac{3\alpha\mu_\mathcal{D}}{R}(f(\theta^1) - f(\theta^*)), \frac{L\alpha^2}{2} \right)$.

We deduce finally that $\delta_{t+1} \leq \frac{a}{t+1}$. Therefore, we get: $\forall T \geq \left[\frac{\alpha\mu_\mathcal{D}}{R}\right] + 1,\ \mathbb{E}[f(\theta^T)] - f(\theta^*) \leq \frac{a}{T}$, and using equation 6, we deduce that:

$$\forall T \geq 1,\ \mathbb{E}[f(\theta^T)] - f(\theta^*) \leq \frac{a}{T}.$$

In particular, if $\mu_\mathcal{D}$ is proportional to $\frac{1}{\sqrt{d}}$, then by taking $\alpha = \frac{2R}{\mu_\mathcal{D}}$, we have:

$$a = \max(\frac{3\alpha\mu_\mathcal{D}}{R}(f(\theta^1) - f(\theta^*)), \frac{L\alpha^2}{2(\frac{\alpha\mu_\mathcal{D}}{R} - 1)}) = \max(6(f(\theta^1) - f(\theta^*)), \frac{L\frac{4R^2}{\mu_\mathcal{D}^2}}{2}) = O(d),$$

therefore $\mathbb{E}[f(\theta^T)] - f(\theta^*) = O(\frac{d}{T})$.  $\square$

*Proof of Theorem 5.* By employing the first part of the proof of Theorem 4, we have that:

$$\forall t \geq 1,\ \mathbb{E}[f(\theta^{t+1}) - f(\theta^*) \mid \theta^t] \leq \left(1 - \alpha_t \frac{\mu_\mathcal{D}}{R}\right)\left(f(\theta^t) - f(\theta^*)\right) + \frac{L\alpha_t^2}{2}$$

Using Lemma 8, we deduce that when $\alpha_t = O(\frac{1}{t^{1-\theta}})$ with $\theta \in (0, \frac{1}{2})$, we get:

$$\forall \epsilon \in (2\theta, 1),\ f(\theta^T) - f(\theta^*) = o(\frac{1}{T^{1-\epsilon}})\ \text{a.s.}$$

$\square$

*Proof of Lemma 2.* Let $t \geq 1$. Using the smoothness property in equation 1, we have:

$$\begin{cases} f(\theta^t + \alpha_t s_t) \leq f(\theta^t) + \alpha_t\langle\nabla f(\theta^t), s_t\rangle + \frac{L}{2}\alpha_t^2\|s_t\|^2 \\ f(\theta^t - \alpha_t s_t) \leq f(\theta^t) - \alpha_t\langle\nabla f(\theta^t), s_t\rangle + \frac{L}{2}\alpha_t^2\|s_t\|^2 \end{cases} .$$

Then $f(\theta^{t+1}) \leq f(\theta^t) - \alpha_t|\langle\nabla f(\theta^t), s_t\rangle| + \frac{L}{2}\alpha_t^2\|s_t\|^2$. By replacing $\alpha_t$ by its expression, and using Assumption 6, we have:

$$f(\theta^{t+1}) \leq f(\theta^t) - \frac{|f(\theta^t + h^{-t}s_t) - f(\theta^t)|}{Lh^{-t}}|\langle\nabla f(\theta^t), s_t\rangle| + \frac{L}{2}\left(\frac{f(\theta^t + h^{-t}s_t) - f(\theta^t)}{Lh^{-t}}\right)^2\ \text{a.s.}$$

$$\leq f(\theta^t) - \frac{|f(\theta^t + h^{-t}s_t) - f(\theta^t)|}{Lh^{-t}}|\langle\nabla f(\theta^t), s_t\rangle|$$

$$+ \frac{L}{2}\left(\frac{|f(\theta^t + h^{-t}s_t) - f(\theta^t)| - |\langle\nabla f(\theta^t), h^{-t}s_t\rangle|}{Lh^{-t}}\right)^2$$

$$+ \frac{|f(\theta^t + h^{-t}s_t) - f(\theta^t)|\,|\langle\nabla f(\theta^t), h^{-t}s_t\rangle|}{Lh^{-2t}} - \frac{|\langle\nabla f(\theta^t), h^{-t}s_t\rangle|^2}{2Lh^{-2t}}\ \text{a.s.}$$

$$\leq f(\theta^t) - \frac{|\langle\nabla f(\theta^t), s_t\rangle|^2}{2L} + \frac{L}{2}\left(\frac{|f(\theta^t + h^{-t}s_t) - f(\theta^t)| - |\langle\nabla f(\theta^t), h^{-t}s_t\rangle|}{Lh^{-t}}\right)^2\ \text{a.s.}$$

$$\leq f(\theta^t) - \frac{|\langle\nabla f(\theta^t), s_t\rangle|^2}{2L} + \frac{L}{2}\left(\frac{|f(\theta^t + h^{-t}s_t) - f(\theta^t) - \langle\nabla f(\theta^t), h^{-t}s_t\rangle|}{Lh^{-t}}\right)^2\ \text{a.s.}$$

$$\leq f(\theta^t) - \frac{|\langle\nabla f(\theta^t), s_t\rangle|^2}{2L} + \frac{L}{2}\left(\frac{\frac{Lh^{-2t}\|s_t\|^2}{2}}{Lh^{-t}}\right)^2\ \text{a.s. (using property equation 1)}$$

$$\leq f(\theta^t) - \frac{|\langle\nabla f(\theta^t), s_t\rangle|^2}{2L} + \frac{L}{8}h^{-2t}\ \text{a.s.}$$

We conclude that: $\forall t \geq 1, \; f(\theta^{t+1}) \leq f(\theta^t) - \frac{|\langle \nabla f(\theta^t), s_t \rangle|^2}{2L} + \frac{L}{8} h^{-2t}$ a.s. $\qquad\qquad$ □

*Proof of Lemma 3.* Let $t \geq 1$. By Lemma 2, we have:

$$f(\theta^{t+1}) \leq f(\theta^t) - \frac{\left|\langle \nabla f(\theta^t), s_t \rangle\right|^2}{2L} + \frac{L h^{-2t}}{8} \quad \text{a.s.}$$

Using the tower property:

$$\mathbb{E}\left[\left|\langle \nabla f(\theta^t), s_t \rangle\right|^2\right] = \mathbb{E}\left[\mathbb{E}_{s_t \sim \mathcal{D}}\left[\left|\langle \nabla f(\theta^t), s_t \rangle\right|^2 \mid \theta^t\right]\right]$$

$$\underbrace{\geq}_{\text{Jensen Inequality}} \mathbb{E}\left[\left(\mathbb{E}_{s_t \sim \mathcal{D}}\left[\left|\langle \nabla f(\theta^t), s_t \rangle\right| \mid \theta^t\right]\right)^2\right]$$

$$\underbrace{\geq}_{\text{Assumption 5}} \mu_{\mathcal{D}}^2 \mathbb{E}\left[\left\|\nabla f(\theta^t)\right\|_2^2\right].$$

It holds that: $\mathbb{E}\left[f\left(\theta^{t+1}\right)\right] - f(\theta^*) \leq \mathbb{E}\left[f\left(\theta^t\right) - f(\theta^*)\right] - \frac{\mu_{\mathcal{D}}^2}{2L}\mathbb{E}\left[\left\|\nabla f\left(\theta^t\right)\right\|_2^2\right] + \frac{L h^{-2t}}{8}.$

By Assumption 4, we have $\left\|\nabla f\left(\theta^t\right)\right\|_2^2 \geq 2\mu\left(f\left(\theta^t\right) - f(\theta^*)\right)$, then:

$$\mathbb{E}\left[f\left(\theta^{t+1}\right) - f(\theta^*)\right] \leq \left(1 - \frac{\mu_{\mathcal{D}}^2 \mu}{L}\right)\mathbb{E}\left[f\left(\theta^t\right) - f(\theta^*)\right] + \frac{L h^{-2t}}{8}.$$

Thus by induction we obtain:

$$\forall T \geq 2, \; \mathbb{E}[f(\theta^T) - f(\theta^*)] \leq \left(1 - \frac{\mu_{\mathcal{D}}^2 \mu}{L}\right)^{T-1}[f(\theta^1) - f(\theta^*)] + \frac{L}{8}\sum_{i=1}^{T-1}\left(1 - \frac{\mu_{\mathcal{D}}^2 \mu}{L}\right)^{T-1-i} h^{-2i}.$$

$\qquad\qquad$ □

*Proof of Theorem 6.* Since $h \in \left(\frac{1}{\sqrt{1 - \frac{\mu_{\mathcal{D}}^2 \mu}{L}}}, \infty\right)$, it holds that: $\forall T \geq 2, \; 1 - \frac{1}{h^2\left(1 - \frac{\mu_{\mathcal{D}}^2 \mu}{L}\right)} > 0.$

Then, using Lemma 3, we get:

$$\forall T \geq 2, \; \mathbb{E}[f(\theta^T) - f(\theta^*)] \leq \left(1 - \frac{\mu_{\mathcal{D}}^2 \mu}{L}\right)^{T-1}[f(\theta^1) - f(\theta^*)] + \frac{L}{8h^2}\frac{\left(1 - \frac{\mu_{\mathcal{D}}^2 \mu}{L}\right)^{T-2}}{1 - \frac{1}{h^2\left(1 - \frac{\mu_{\mathcal{D}}^2 \mu}{L}\right)}}$$

$$\leq \left(1 - \frac{\mu_{\mathcal{D}}^2 \mu}{L}\right)^{T-1}\left[f(\theta^1) - f(\theta^*) + \frac{L}{8}\frac{1}{h^2\left(1 - \frac{\mu_{\mathcal{D}}^2 \mu}{L}\right) - 1}\right],$$

which gives the desired inequality.

In the particular case where $\mu_{\mathcal{D}} = \frac{K}{\sqrt{d}}$, by replacing $h$ and $\mu_{\mathcal{D}}$ by their formulas, we obtain the desired rate $O\left(\left(1 - \frac{\mu K^2}{dL}\right)^T\right).$ $\qquad\qquad$ □

*Proof of Theorem 7.* Let $s \in (0, 1)$, we consider $a = 1 - s\frac{\mu_{\mathcal{D}}^2 \mu}{L}$. By multiplying the inequality (3) by $a^{-T}$, we get that for all $T \geq 2$:

$$\mathbb{E}[a^{-T}\left(f(\theta^T) - f(\theta^*)\right)] \leq \left(a^{-1} - \frac{a^{-1}\mu_{\mathcal{D}}^2 \mu}{L}\right)^{T-1}\left[a^{-1}\left(f(\theta^1) - f(\theta^*)\right) + \frac{a^{-1}L}{8}\frac{1}{h^2\left(1 - \frac{\mu_{\mathcal{D}}^2 \mu}{L}\right) - 1}\right].$$

As $a^{-1} - \frac{a^{-1}\mu_\mathcal{D}^2\mu}{L} = \frac{1-\frac{\mu_\mathcal{D}^2\mu}{L}}{1-s\frac{\mu_\mathcal{D}^2\mu}{L}} \in (0,1)$ , it holds that:

$$\mathbb{E}[\sum_{T=2}^{\infty} a^{-T} \left( f(\theta^T) - f(\theta^*) \right)] = \sum_{T=2}^{\infty} \mathbb{E}[a^{-T} \left( f(\theta^T) - f(\theta^*) \right)] < \infty.$$

Therefore: $\sum_{T=2}^{\infty} a^{-T} \left( f(\theta^T) - f(\theta^*) \right) < \infty$ a.s., and we conclude that:

$$f(\theta^T) - f(\theta^*) = o\left( a^T \right) \quad \text{a.s.}$$

$\square$

