# OpenReview forum: "On the Almost Sure Convergence of the Stochastic Three Points Algorithm"
_ICLR.cc/2025/Conference — ICLR 2025 Poster_

### Official Review · Reviewer_tFkj · 2024-10-29

**Soundness:** 4
**Presentation:** 4
**Contribution:** 3
**Rating:** 6
**Confidence:** 3

**Summary:**

Convergence analysis of the stochastic three points (STP) algorithm is studied. The rates for different classes of functions are shown. A almost sure convergence rate arbitrarily close to $O(1/\sqrt{T})$ is shown for smooth functions.
And almost sure convergence rate arbitrarily close to $o(1/T)$ and in expectation a rate of $O(d/T)$ is shown for smooth convex functions. For the class of strong convex functions, a rate of $O((1-\mu/dL)^T)$ in expectation and a rate arbitrarily close to $o((1-\mu/dL)^T)$ for almost sure convergence is shown.

**Strengths:**

Paper is well-written. Convergence rates of the STP algorithm for smooth, smooth convex and smooth strongly convex functions are shown.

**Weaknesses:**

The results are incremental. How does the convergence rate of STP compare with other methods both in expectation and almost sure. For example with RGF or GLD which is compared in the experiments ? This could be discussed or given as a table. What about the optimal convergence rate in each case ?  A brief discussion about the advantages of STP algorithm in the introduction would be helpful.

**Questions:**

1. Please add a table comparing the convergence rates (both in expectation and almost sure) of STP, RGF, and GLD for each function class studied. Also please discuss how STP's rates compare to known optimal rates for each case. This is important as it would provide valuable context for the significance of the STP results.

2. Please add a paragraph or two in the introduction highlighting the key advantages of STP over other zeroth-order methods, such as its linear dependence on dimension compared to quadratic dependence for deterministic direct search methods. This would help readers better understand the importance of studying STP's convergence properties.

---

> ### Author Response · Authors · 2024-11-21
> **Reply to Reviewer tFkj**
>
> We thank the reviewer for the positive feedback and helpful suggestions. Below, we address the points raised by the reviewer:
>
> In the revised manuscript, we enhanced the introduction to emphasize the theoretical significance of the STP algorithm by comparing it to other zeroth-order methods, including RGF and GLD (lines 62-85 and 104-120). We mentioned (line 84) that the complexity bounds of the STP algorithm match the best-known ones in the literature on zeroth-order methods. However, to the best of our knowledge, unlike first-order methods, no lower bounds have been established for zeroth-order methods to certify the optimality of the STP convergence rates.

---

### Official Review · Reviewer_dTaK · 2024-11-01

**Soundness:** 3
**Presentation:** 2
**Contribution:** 2
**Rating:** 6
**Confidence:** 3

**Summary:**

This paper establishes the almost sure (and last iterate) convergence of the stochastic three-point algorithm for zeroth-order optimization. The paper considers smooth nonconvex, convex, and strongly convex settings and gets the results under three different settings, respectively. Numerical experiments demonstrate the performance of the proposed method.

**Strengths:**

The paper is well-written, with technical results clearly presented and explained. Also, different settings are considered in the paper.

**Weaknesses:**

My major concern is the technical novelty of the paper.

1. Insufficient technical contribution.

   Although the paper considers multiple settings and gets almost-sure convergence results, the techniques seem to be a combination of [1] and other papers that establish almost-sure convergence of SGD in different settings. Especially the results in section 3 and 4 look less surprising, since most of them are obtained by verifying some conditions and invoking an existing almost-sure convergence result from the SGD literature.

2. Presentation issues.

   The paper contains many technical results, either proven in this paper or cited from other papers. However, there is not sufficient explanation after each main result. This may make the results less accessible to the readers.

Overall I think this is a solid work, and I would like the authors to further clarify the technical contributions in this paper.

**Questions:**

1. In the nonconvex case (Theorem 1), lower-boundedness (A2) is not assumed. But on line 624 in the appendix, the authors use the assumption that $f$ is bounded from below.
2. A7 seems to be an artifact of analysis. Could you provide more intuitions behind it?
3. Results in section 4 require the initial point to lie in the sublevel set. Here the cost of finding such an initial point is not discussed. Could you elaborate more on this?

**Minor issues**

1. Line 139

   There seems to be an additional 2 in the complexity.

2. Line 146

   The Assump column does not match all the results in the paper (e.g., Theorem 7 in section 5 mentions "Assumption 1 and 4 to 7"). Also, assumption 7 does not appear in the table.

3. Line 146, 533

   Please put the caption of the table at the top.

4. Line 204

   The statement of the examples seems incorrect and inconsistent with [1]. The covariance matrix in [1] is $I_d / d$  instead of $I_d$.

   For any arbitrary => For any.

5. Line 318, 372

   $\ast$ and $^\star$ are inconsistent.

6. Line 890

   $m$ is undefined.

**References**

[1] Bergou, E. H., Gorbunov, E., & Richtarik, P. (2020). Stochastic three points method for unconstrained smooth minimization. *SIAM Journal on Optimization*, *30*(4), 2726-2749.

---

> ### Author Response · Authors · 2024-11-21
> **Reply to Reviewer dTaK**
>
> We thank the reviewer for the constructive feedback and helpful suggestions. Below, we address the points raised by the reviewer:
>
> Concerning the technical contribution; we added two novel result/proof to make our contributions less dependent on existing results from the SGD literature:
> 1. In the revised version, we present a new proof for the result (Theorem 1) regarding the rate of the best gradient iterate. The new proof is based on a new result (mentioned in Lemma 5 of our revised version). Our proof of Lemma 5 is significantly simpler than the approach used, in the SGD litterature, by Liu et al., which is based on (Lemma 2, [1]) that has a highly technical proof and depends on (Proposition 1, [1]), a classical supermartingale convergence theorem first showed by Robbins and Siegmund [2].
>
> 2. In the revised version, the convergence results of the last iterate (Theorems 2 and 3 of our paper), whether in expectation or almost surely, are based on Lemma 7 (of the revised version). This lemma was first introduced in (Proposition 2, [3]) and later revisited in (Lemma A.5, [4]), where another proof was provided. In contrast, we present a new and simpler proof that is more straightforward than those given in these earlier works.
>
> Concerning the presentation issues, we added explanations to improve the presentation of the results.
>
> Question 1: Yes on line 624 in the appendix (of the initial version of our paper), we used the assumption that $f$ is bounded from below. This assumption was made for all the sections of the paper (mentionned in line 168 in the old version, corresponding to line 201 in the new version). However, this assumption is different from assumption (A2) that assumes the existence of the minimum and that we are not using in the proof of Theorem 1.
>
> Question 2: We changed assumption A7 to the more simpler assumption $\mu_D<1$ which is sufficent for theorems 6 and 7 (since $\mu/L\leq 1$). This new assumption is satisfied by classical distributions (see Lemma 3.4 of [5]).
>
> Question 3: Indeed, in the convex case, we require the initial point to lie within a bounded sublevel set. However, we do not know how to guarantee that the initial point lies in such a sublevel set. Note that in the convex case, when the search directions are chosen from the canonical vectors of $\mathbb{R}^d$, the STP algorithm can be viewed as a zeroth-order version of the randomized coordinate gradient descent algorithm, where the same assumption is needed (see Assumption 1 in [6]).
>
> We also thank the reviewer for the other remarks (given in the "Minor issues" section) that we addressed in the new version of the paper.
>
> [1] Jun Liu and Ye Yuan. On almost sure convergence rates of stochastic gradient methods. In Conference on Learning Theory, pp. 2963–2983. PMLR, 2022.
>
> [2] Herbert Robbins and David Siegmund. A convergence theorem for non negative almost supermartingales and some applications. In Optimizing Methods in Statistics, pages 233–257. Elsevier, 1971.
>
> [3] Ya I Alber, Alfredo N Iusem, and Mikhail V Solodov. On the projected subgradient method for nonsmooth convex optimization in a hilbert space. Mathematical Programming, 81:23–35, 1998.
>
> [4] Julien Mairal. Stochastic majorization-minimization algorithms for large-scale optimization. Advances in Neural Information Processing Systems, 26, 2013.
>
> [5] El Houcine Bergou, Eduard Gorbunov, and Peter Richtarik. Stochastic three points method for unconstrained smooth minimization. SIAM Journal on Optimization, 30(4):2726–2749, 2020.
>
> [6] Stephen J Wright. Coordinate descent algorithms. Mathematical programming, 151(1):3–34, 2015.

---

> > ### Comment · Reviewer_dTaK · 2024-11-22
> >
> > Thank you for the response. The authors address some of my concerns. Based on the revision, here are some additional comments.
> >
> > 1. In the statement of Theorem 1, should it be $T \rightarrow +\infty$? Also, on line 487, 744, and 749, it should be $\min_{t \leq T} \\|\nabla f(\theta^t)\\|$ instead of $\min_{t \leq T} \\|\nabla f(\theta^T)\\|$. And the paper seems to be inconsistent with $\sum_{t \geq 1}$ and  $\sum_{t=1}^{+\infty}$ in many places. I recommend that the authors proofread the paper carefully. Otherwise, the paper will not be accessible to the readers even if it contains some solid results.
> > 2. In the updated experiments, it is hard to compare the algorithms since the figures have different ranges in the y-axis (also x-axis in the solution time comparison). Moreover, RGF seems to outperform STP in both convergence and solution time.
> > 3. The complexity of finding an initial point in a given sublevel set is not sufficiently discussed. To make the algorithm practically useful, I don't think it is sufficient to simply mention that "the assumption is made in literature".

---

> > > ### Author Response · Authors · 2024-11-23
> > >
> > > Thank you for your response. We greatly appreciate the time and effort you dedicated to reviewing the manuscript, as well as your additional comments that allowed to improve further our paper. Below, we address each of your points:
> > >
> > > 1. Yes, all the limits considered in the paper are with respect to $T\to +\infty$. We have revised the paper and addressed the different typos, in particular the inconsistency mentioned.
> > >
> > > 2. We revised the experiments using a higher dimension, $d = 500$, and we have standardized the scales of the y-axis, as well as the x-axis for the plots vs Time. In these updated experiments, where $d = 500$, the STP slightly outperforms the RGF algorithm. In fact, we observed experimentally that STP improves over RGF when the dimensionality increases, even if theoretically they have the same dependence on the dimension.
> > >
> > > 3. We thank you especially for this comment. Indeed, we missed an existing result in the convex analysis literature, which states the following: for a closed, proper convex function $f$, if one of its sublevel sets is bounded, then all its sublevel sets are bounded (see Corollary 8.7.1 in [1]). In our setting of section 4, $f$ is a real-valued, continuous, and convex function, and thus a closed, proper convex function. Therefore, under our Assumption 3, where we assume the existence of a vector whose sublevel set is bounded, the conditions of that corollary are satisfied. This guarantees that the sublevel set of any initial vector $\theta^1$ is bounded, ensuring the algorithm's correctness from any starting point.
> > > We discussed this in the first part of section 4 (line 333-354 of the new version) and we removed Remarks 5 and 6 related to the issue of choosing the initial point $\theta^1$, as they are no longer needed.
> > >
> > >
> > > [1] Ralph Tyrell Rockafellar. Convex analysis:(pms-28). 2015.

---

> > > > ### Comment · Reviewer_dTaK · 2024-11-26
> > > >
> > > > Thank you for the response. The authors addressed some of my major concerns, and overall, I increased my score to 6.
> > > > I still noticed some stylistic issues (e.g., there is space before ":"). Please fix them in the next revision.

---

> ### Author Response · Authors · 2024-11-26
>
> We thank you for your response, time, and re-evaluation of the score. We have addressed the stylistic issues in the new uploaded version of the paper.

---

### Official Review · Reviewer_3UjN · 2024-11-04

**Soundness:** 3
**Presentation:** 3
**Contribution:** 3
**Rating:** 6
**Confidence:** 3

**Summary:**

This paper analyzes almost-sure convergence for the derivative-free stochastic three points (STP) algorithm in (Bergou et al., 2020), where only the convergence in expectation was studied. The convergence in expectation does not guarantee convergence of individual instance of trajectories, which is why, for example, extensive research on almost-sure convergence has been conducted for SGD. This paper's study of STP, therefore, aligns with recent theoretical advancements in SGD. In particular, this paper provides the almost-sure convergence results for three standard classes of functions, as summarized in Table 1.

**Strengths:**

This paper provides the first almost-sure convergence analysis of the STP method for three standard classes of functions, a non-trivial achievement. This analysis guarantees convergence of each trajectory instance, making it valuable both theoretically and practically.

**Weaknesses:**

- Motivation: The focus on STP in this paper is not well motivated in the abstract and introduction, aside from mentiong its low per-iteration complexity. Is the STP widely used in practice? Given its simplicity and strong theoretical guarantees, I cannot see why this would not be considered by practitioners. I suggest that the authors further elaborate on the theoretical and practical importance of studying the STP.

- Comparison to other existing derivative-free methods: Although this paper focuses on the STP, it would have been helpful to include comparisons or discussions of the almost-sure convergence (rate) analyses for other existing derivative-free methods, if available. Without these, it is difficult to locate this work within the broader literature.

- Experiment: What is the purpose of this experiment? It seems quite orthogonal to the paper's theoretical contributions. It looks like it was added to appeal to practitioner reviewers, but I believe it may ultimately satisfy neither theory nor practitioner reviewers. I suggest revising the experiment section to better align with the paper's main contributions.

**Questions:**

- Lines 111-112: Although it is straightforward, please explicitly state the consequence of the result.
- Line 124: $\theta$ denotes both the iterates and the step size parameter, and I suggest using other letter for the step size parameter.
- Remark 2: This seems redundant and I suggest removing it.
- Lemma 4: This does not seem necessary here, which is only used in the proof in the Appendix. I suggest moving it to Appendix.
- Remarks 7,8 and 10: They discuss the condition that Assumption 3 can be satisfied. I believe that they can be simply discussed right after Assumption 3, or can be more clearly and briefly stated in the remarks.

---

> ### Author Response · Authors · 2024-11-21
> **Reply to Reviewer 3UjN**
>
> We thank the reviewer for the constructive feedback and the valuable comments. We address the concerns and questions raised by the reviewer below:
> 1. Motivation:
> In the revised manuscript, we updated the introduction to better highlight the theoretical and practical importance of the STP algorithm.
> For the theoretical importance, we added two paragraphs (lines 62-85). For the practical part, we included a new paragraph (lines 86-98).
>
> 2. Comparison to other existing derivative-free methods:
> We add comparison to other derivative-free methods in termes of convergence in expectation (lines 62-85) and in termes of the almost sure convergence (lines 104-120).
>
> 3. Experiment:
> We revised the experiment section in order to illustrate better the almost sure convergence of the STP algorithm.
>
> We also thank the reviewer for the other remarks (given in the "Questions" section) that we addressed in the new version of the paper.

---

> > ### Comment · Reviewer_3UjN · 2024-11-27
> >
> > Thank you for your response, and I apologize for the delay in getting back to you. I appreciate the authors' efforts to address the feedback and improve the paper. After carefully reviewing the other comments and the authors' response, I have decided to increase my score to 6, recognizing the value of studying the almost-sure convergence of zeroth-order methods.
> >
> > I would like to explain my reasoning for not raising the score further. While the paper offers a meaningful contribution, it feels somewhat specific to a particular zeroth-order method and relies significantly on existing results for almost-sure convergence in SGD, despite the introduction of a simpler analysis during the rebuttal. As a result, the broader impact of the work seems somewhat limited.
> >
> > That said, given the experimental results, I am curious whether the observed almost-sure convergence could be more universal and extend to other zeroth-order methods such as GLD and RGF. Exploring these directions could further enhance the significance and applicability of the work.

---

> > > ### Author Response · Authors · 2024-11-27
> > >
> > > We thank you for your response, time, and re-evaluation of the score. We sincerely appreciate your valuable comments, which have greatly contributed to improving our manuscript. We also thank you for the proposed exploration directions. We will certainly explore the possibility of extending the almost-sure convergence results to other zeroth-order methods.

---

### Author Response · Authors · 2024-11-21
**General Response**

We thank all the reviewers for their constructive feedback. Your comments and questions were invaluable in improving our paper. We have uploaded a revised version of our work taking into account your suggestions. Additionally, the supplementary material includes this updated version with all changes clearly highlighted in red.

---

### Meta-Review · Area_Chair_SRAK · 2024-12-19

**Metareview:**

The paper studies the almost sure convergence of a specific algorithm for zeroth-order optimization called the stochastic three points (STP) algorithm, in three main settings of smooth optimization: nonconvex, convex, and strongly convex. The main result are the rates of almost sure convergence for all three settings, significantly strengthening the previous results that were only applicable in expectation. On a technical level, the paper combines results for almost sure convergence of SGD and prior "in expectation" results for STP, which makes it somewhat limited. Nevertheless, the authors provided a more elegant analysis in the rebuttal and there was ultimately a consensus that the paper should be accepted.

A small comment: I found the statements like "at a rate arbitrarily close to <little oh of something>" confusing. Something is either "little oh" or "Big Oh." It is unclear what "arbitrarily close" to it even means. Please make sure to state the results more clearly in the abstract.

**Additional Comments On Reviewer Discussion:**

The main concerns raised in the reviews were with respect to the novelty/relationship to prior work and strength of contributions. The authors revised the paper in the rebuttal to include a new, simpler analysis. Although this seems to have convinced the reviewers, the fact that the results could be (and were) obtained as a combination of prior techniques + the fact that the paper is somewhat niche since if considers a specific prior algorithm make the contributions somewhat limited. Nevertheless, all reviewers were ultimately convinced that the paper should be accepted (with limited enthusiasm).

---

### Decision · Program_Chairs · 2025-01-22

Accept (Poster)